# Predictive Factors of Response to Neoadjuvant Chemotherapy (NACT) and Immune Checkpoint Inhibitors in Early-Stage Triple-Negative Breast Cancer Patients (TNBC)

**DOI:** 10.3390/curroncol32070387

**Published:** 2025-07-04

**Authors:** Khashayar Yazdanpanah Ardakani, Francesca Fulvia Pepe, Serena Capici, Thoma Dario Clementi, Marina Elena Cazzaniga

**Affiliations:** 1School of Medicine and Surgery, University of Milano-Bicocca, 20900 Monza, Italy; k.yazdanpanaharda@campus.unimib.it (K.Y.A.); t.clementi@campus.unimib.it (T.D.C.); 2Phase 1 Research Center, Fondazione IRCC San Gerardo dei Tintori, 20090 Monza, Italy; francescafulvia.pepe@irccs-sangerardo.it (F.F.P.); serena.capici@irccs-sangerardo.it (S.C.)

**Keywords:** triple-negative breast cancer, circulating tumor cells, circulating tumor DNA, TNBC, immunotherapy

## Abstract

Triple-negative breast cancer (TNBC) is a type of breast cancer very aggressive. Scientific efforts are undergoing trying to better understand how each person’s cancer might respond to treatment, before starting therapy. Currently, the main treatment combines chemotherapy with drugs that help the immune system fight cancer. We looked at several factors that might help response prediction, like immune cells inside the tumor, cancer cells or DNA detected in patients’ blood, a gene called p53 and a marker called Ki-67 which represents cancer growing. By studying these factors, the goal is to personalize treatment and improve outcomes for people with TNBC.

## 1. Introduction

The 2020 GLOBOCAN database reported approximately 2.3 million new cases of breast cancer (BC) globally, representing the most prevalent malignant tumor among women worldwide. There are notable discrepancies in cancer epidemiology across European countries, including considerable variations in cancer incidence and mortality [1]. Triple-negative breast cancer (TNBC) accounts for 15–25% of all cases of breast cancer [2]. African American and Hispanic women are found to be at higher risk of developing TNBC, with poorer prognosis in African American women. This type of breast tumor has a higher probability of developing in women at a young age; similarly, older women might be prone to developing TNBC due to cumulative changes that can occur in breast cancer gene 1 (BRCA1) and breast cancer gene 2 (BRCA2) [3]. There were an estimated 2,088,849 reported cases of TNBC in 2018, with a steady increase in the incidence rate [4,5].

Neoadjuvant chemotherapy (NAC) is a common treatment for patients with newly diagnosed TNBC, as it allows for pathological response-guided adjuvant therapy [6]. In comparison with non-TNBC disease, TNBC has been linked to higher rates of recurrence and poorer overall survival (OS) and progression-free survival (PFS). Consequently, there is a continued need to identify prognostic and predictive biomarkers for this population to enhance treatment responses [7]. Very recently, the availability of immune checkpoint inhibitors, especially Pembrolizumab, has led to the wider and more effective treatment of TNBC patients.

This review investigates the precision of the most recent predictive factors regarding responses in early-stage TNBC patients. Thus, in the future, we may be able to develop a clear understanding of the patient’s prognosis at the outset of treatment and the possibility of adapting the course of the treatment accordingly.

## 2. Methods

Our search strategy involved the following keywords in the PubMed search engine: “triple-negative breast cancer”, “TNBC”, “tumor infiltrating lymphocytes”, “TILs”, “liquid biopsy”, “ctDNA”, “CTC”, “p53 mutation”, “neutrophil to lymphocyte ratio”,” NLR”, “platelets to lymphocytes ratio”, “PLR”, “predictive factors”, and “prognostic factors”. We included studies focusing on human cohort studies, randomized/non-randomized clinical trials, meta-analyses, and review articles.

Initially, we included 77 studies; after applying the exclusion criteria, 42 studies were considered for the purposes of this review. Some studies that did not provide direct information relevant to our topic, such as those primarily focusing on other types of breast cancer without a meaningful reference to TNBC, were included only if they contributed valuable context or mechanistic insight. Our original focus was on the literature published in the last 10 years. However, taking into consideration the information and valuable explanations about our chosen subjects, we subsequently expanded the studies included beyond the initial determined timeframe. In the later stages of the research, in order to achieve a clearer and more thorough understanding of the topic, we decided to briefly describe the results in terms of pCR and the survival outcomes of immune checkpoint inhibitors (ICIs), as the introduction of these agents in the neoadjuvant treatment of TNBC patients could, in the near future, open up a new window on different biomarkers. During the later revisions, the role of Ki-67 as a potential marker for response to treatment in TNBC patients was investigated, resulting in the inclusion of seven additional studies. In terms of the results, in order to obtain a comprehensive conclusion, and by summarizing the results alongside a brief explanation of the underlying mechanisms, seven additional studies were considered. A discussion of future directions is also included in this narrative review as the final section. The goal of this section is to demonstrate some of the latest developments in certain types of solid tumors and provide researchers with new ideas regarding the way in which they can focus their research.

## 3. Results

For the purposes of this review and for the convenience of the reader, while at the same time providing the most comprehensive description of the data we were able to retrieve, biomarker-related results have been split into two relevant sections: those obtained from (1) tumor tissue and (2) those derived from liquid biopsy.

### 3.1. Tissue-Based Biomarkers

#### 3.1.1. Tumor-Infiltrating Lymphocyte (TILs) and Other Immune Factors

Tumor-infiltrating lymphocytes (TILs) are defined as lymphocytes that migrate from the blood into the mass of cancerous cells [8]. The significance of these immune cells’ infiltration is regarded as particularly important in the context of treating immune checkpoint inhibitors [9,10].

Two types of lymphocyte infiltration can be present at the site of a tumor, namely stromal tumor infiltrating lymphocytes (sTILs) and intra-tumoral tumor infiltrating lymphocytes (it-TILs). Denkert et al., 2009 [11] studied the association between TILs and pCR in breast cancer patients’ samples over two studies. In one GeparDuo trial involving breast cancer patients at stages T2-3, N0-2, and M0 of the disease, all patients received NACT. In another GeparTrio trial, with stages T2-4, N0-3, and M0 present, all patients again received NACT. The results indicated that in lymphocyte-predominant tumors, the pCR rates were 42% and 40% in the training cohort and validation cohort groups, respectively. In contrast, in the tumors with no lymphocyte infiltration, the pCR rates were 3% and 7%, respectively [11].

Dieci et al. investigated the effects of sTILs and it-TILs on outcomes in 278 TNBC patients with residual disease after surgery; the authors showed that higher TIL rates were associated with an absence of axillary nodal disease and smaller tumor size at surgery. After a median follow-up time of 6.3 years, it was observed that, for each 10% increase in sTILs, the risk of metastasis and death decreased by 21%. Similarly, an increase in it-TILs was associated with a decrease in metastasis development and death: for each 10% increase in it-TILs, a 22% reduction in the risk of metastasis and 23% reduction in the death rate were observed [12].

Yuka Asano et al. evaluated the association of pCR and TIL rate in the surgical specimens of 177 patients with stage IIA, IIB or IIIA TNBC or HER2 + breast cancer (BC) receiving neoadjuvant chemotherapy (NACT). They concluded that in both subtypes of breast cancer considered, patients with higher TIL levels had a higher rate of pCR in comparison to patients with lower TIL levels. They also investigated TIL levels after disease recurrence and observed a reduction in TIL levels in the recurrent disease after a successful course of NACT. In their study, out of 177 breast cancer patients, 61 were diagnosed with TNBC, of whom 48 patients had high TIL levels, and 13 were determined to have low levels of TILs; 26 patients with high TILs reached pCR, while the pCR rate was 2 out of 13 patients in the group with low TILs [13]. A meta-analysis which included 37 studies investigated the potential correlation among pCR rate, disease-free survival (DFS), and overall survival (OS) in TNBC patients with high and low TIL levels. The results confirmed that patients with higher TIL levels had a higher rate of pathologic complete response (pCR). In terms of DFS and OS, patients with higher levels of TILs had better outcomes. Additionally, they compared different levels of CD4+ and CD8+ T-cells. Higher CD4+ infiltration was associated with better OS and DFS; while patients with CD8+ T-cells only had better DFS, the same was true for higher levels of FOXP3+ TIL levels [14]. Finally, a pooled analysis of six German Breast Group (GBG) studies (906 TNBC patients) confirmed the correlation between sTILs and pCR [15].

Table 1 summarizes the main data available for TILs.

In short, TILs could be considered a strong positive predictive factor for response to NACT, especially in terms of stromal proportion, and a promising tool as a prognostic biomarker.

#### 3.1.2. Protein 53 (p53) Mutation

Some studies have reported that tumors with a tumor protein 53 (TP53) mutation are able to achieve higher pCR rates. The underlying mechanism supporting this is the inability of the cancer cells to repair their DNA damage, leading to cell death [16]. The role of TP53 mutation as a potential biomarker in TNBC is contradictory, as some studies have demonstrated the high p53 mutation burden having a negative impact on the prognosis of TNBC, while others have shown contrary findings.

Pan et al. tested 156 patients with TNBC for the presence of p53 mutation, finding that p53 positivity was correlated with higher tumor grade and worse prognosis [17].

Another randomized clinical trial investigated the roles of serum lipid levels, TP-53 mutational status, and Ki-67 based on molecular subtypes in breast cancer, as well as evaluating the tissue of 61TNBC patients to assess the presence or absence of a TP-53 mutation: 49 had a p53 mutation and 12 patients showed wild-type p53. The authors concluded that in the case of the TNBC subtype, p53 status was not found to have a specific impact on the complete tumor response rate [18].

Additionally, in a study published in 2016, it seemed that mutations in TP53 in patients with TNBC were associated with higher levels of TILs in comparison to TP53 mutations in other cancer types, or the same mutation in other subtypes of breast cancer; however, the authors failed to demonstrate any association between TP53 status and pCR in this group of patients. An interesting aspect of this paper that deserves attention is that the authors also studied the link between TP53 status and pCR in TNBC stratified according to the chemotherapy regimen used, though no significant association was found [19]. This finding does, however, contrast with the results reported by Lehman-Che J et al., who hypothesized that TP53 mutation could be considered a predictive factor in a specific chemotherapeutic setting, also concluding that pCR in TP53-mutated patients depends on the type of the chemotherapy [20].

In a meta-analysis, Min-Bin Chen et al. evaluated the association between TP53 mutation status and the response to neoadjuvant chemotherapy in 26 studies with a sample population of 3476 subjects. The final conclusion drawn was that an altered TP53 status seemed to be related to a patient’s positive response to NACT: in this meta-analysis, TP53 status was also associated with overall response (OR) and complete response (CR) to the chemotherapy [21].

It is undeniable that the TP53 gene plays a significant role in different cellular processes; in the studies carried out so far, it is clear that the presence of TP53 mutation could be considered a very promising biomarker. Whether it could be associated with response or prognosis remains to be determined through retrospective analyses or, preferably, in the context of prospective trials.

#### 3.1.3. Ki67

Ki-67 is an important cancer biomarker, determining the proliferative capability of cells. An increase in expression is seen during cellular proliferation, reaching its peak in the G2 stage of cell proliferation; it plays a role in chromosomal stability in the nucleus of the proliferative cells. For this reason, cancer cells that rapidly proliferate express high levels of this protein, which can be detected in tissue samples by immunohistochemistry (IHC) and used to determine a cancer cell’s proliferative ability and aggressiveness [22].

Wang et al. analyzed data collected from 280 stage II or III TNBC patients who received the same neoadjuvant chemotherapy regimen involving weekly doses of paclitaxel and carboplatin. This study demonstrated that patients with better outcomes after neoadjuvant chemotherapy also showed a steeper reduction in their Ki-67 levels, with higher pCR rates in patients who showed higher levels of Ki-67 at baseline [23].

Keam et al. evaluated 105 stage II-III TNBC patients receiving NACT with docetaxel and doxorubicin for three cycles every three weeks. Overall, the pCR rate was 13.3%. Patients were categorized into high and low groups based on their Ki-67 levels, with the cut-off fixed at 10%. These authors observed that only the patients with high Ki-67 levels achieved pCR, compared to none of the patients in the low Ki-67 group. Moreover, high Ki-67 levels were associated with worse outcomes in terms of overall survival and relapse-free survival (RFS) [24].

Wang et al. investigated the prognostic value of Ki-67 and TIL levels on residual disease (RD) after NACT followed by surgery in 109 non-metastatic TNBC patients: they observed that in the group that showed no decrease in Ki-67, the influence of TIL levels on the RD was associated with OS and RFS [25].

Other studies also described an association between Ki-67 levels and response to NACT. Rais et al., in a retrospective study of 187 breast cancer patients, assessed the relationship between Ki-67 before the initiation of the therapy and pCR rate after surgery: 14.3% of the patients had TNBC. The cut-off value in order to consider a patient as having high Ki-67 levels was defined as Ki-67 > 35%. The standard chemotherapy in this study administered to all patients was anthracycline-based, followed by taxanes. The authors described a pCR rate of 40% amongst 75 patients out of 187, of whom 64 patients had Ki-67 levels higher than 35% before the initiation of neoadjuvant therapy, and only 11 patients with Ki-67 < 35% achieved pCR [26].

Masuda et al. evaluated the predictive factors of effectiveness of NACT in 33 TNBC patients; chemotherapy regimens administered included epirubicin, cyclophosphamide, and taxanes. Among patients with Ki-67 equal or greater than 50%, a pCR rate of 50% was achieved, while the pCR rate was only 15% in patients with low Ki-67 [27].

Ki-67, as a biomarker of cellular proliferation capability, is a useful biomarker to evaluate how cells are responding to neoadjuvant treatment by comparing results at first biopsy and after surgery. However, it is likely that Ki-67 cannot be indicative of treatment response just by itself, and needs to be used in a combined modality with other factors [28]. Main data are reported in Table 2.

### 3.2. Liquid Biopsy Biomarkers

Based on the definition proposed by the National Cancer Institute (NCI) a liquid biopsy is a sample of blood or any other body fluid of a patient used to determine the presence of tumor cells, DNA, RNA, or other molecules of cancer cells in the sample taken from the patients, without the need for the invasive procedure of sampling the tissue directly [29].

Based on liquid biopsy, two biomarkers have been studied in order to predict tumor response: the number of circulating tumor cells (CTCs) and the evaluation of fragments of tumor DNA, namely circulating tumor DNA (ctDNA) and cell-free DNA (cfDNA).

#### 3.2.1. Circulating Tumor Cells (CTCs)

Cristofanili et al. demonstrated in 2004 [30] that circulating tumor cells (CTCs) can be used to predict disease progression and prognosis earlier than traditional imaging techniques during the follow-up phase, either after surgery or during treatment for metastatic disease. In their study, the presence of CTCs equal to or more than five in each 7.5 mL of blood was an indicator of a poor overall survival (OS) in metastatic BC patients, but most importantly, CTC count was able to predict disease’s progression or prognosis after 3–4 weeks of treatment initiation in comparison to 8–12 weeks of traditional imaging techniques. The authors also emphasized the fact that levels of CTCs before and, more importantly, after the initial dose of treatment showed valuable predictive information for progression-free survival (PFS) and overall survival [30]. Bidard et al., in 2014 [31], evaluated the clinical validity of CTCs in metastatic breast cancer (mBC). Although their study did not exclusively observe TNBC patients, the results underpinned the idea of CTCs as a predictive factor for determining relapse and the severity of the disease in all groups of breast cancer patients.

In their study, the cut-off for high CTCs was 5 CTCs per 7.5 mL or higher, and a value higher than this cut-off was regarded as a predictive factor for presence of bone and liver metastasis, alongside elevated levels of CEA and CA 15-3 in the blood. This study also pointed out that the number of previous chemotherapy lines could be associated with the levels of CTC, as patients who received several lines of chemotherapy had a lower CTC count. In order to observe the predictive value of the CTC count, the authors constructed a model based on the previous factors and their effects on breast cancer overall survival (OS) and progression-free survival (PFS), after having accounted for the predictive value of the CTC count. Their model for PFS included tumor histological subtype and histological grade, the number of previous lines of anti-hormone therapy, the number of previous lines of chemotherapy, the presence of liver and bone metastasis, and performance status; and the model for OS included tumor histological subtype, the number of previous hormone therapies, performance status, and liver metastasis. The addition of the CTC count to the pathological models resulted in the CTC count demonstrating a high predictive value in terms of OS and PFS; in particular, a CTC count higher than 5 CTCs or higher for each 7.5 mL of the blood sample is a significant negative predictive value (NPV) for PFS and OS.

At the beginning, out of 328 patients with a CTC count higher than 5 CTCs or more per 7.5 mL, those (149 cases) that demonstrated an initial decrease to under 5 CTCs/7.5 mL by weeks 3–5 had longer OS and PFS than patients whose CTC count remained higher than the cut-off value.

In the sample of TNBC patients undergoing PFS analyses, out of the 233 TNBC patients included, 197 showed disease progression, indicating that high or increased levels of CTCs in TNBC patients are an indication of a higher risk of disease progression or relapse [31].

#### 3.2.2. Circulating Tumor DNA (ctDNA) and Cell-Free DNA (cfDNA)

Among the biomarkers obtained through liquid biopsy, ctDNA and cfDNA remain, under ideal circumstances, the most promising.

YH et al. evaluated the predictive role of ctDNA in 33 early-stage TNBC patients within the context of adjuvant setting; the presence of ctDNA was detected in 4 patients and all 4 patients had recurrent disease [32].

Radovich et al. explored ctDNA detection and its association with distant disease-free survival (DDFS), disease-free survival (DFS), and overall survival (OS), in early-stage TNBC in 196 patients receiving neoadjuvant chemotherapy (NACT). They divided their patients into 2 arms, with one group giving the blood sample prior to the treatment on day 1 (arm A), and the other at the routine first visit (arm B). CTC count and ctDNA were measured in both groups. The presence of ctDNA was confirmed in the majority of patients in both groups: 37 out of 57 patients in arm A (65%) and 53 out of 85 for arm B (62%). Positivity for ctDNA was significantly associated with worse outcomes in all three investigated criteria, namely DDFS, DFS, and OS.

Patients who tested positive for ctDNA had significantly worse outcomes than ctDNA-negative patients (HR = 2.99). The 2-year DDFS rate was 56% in ctDNA-positive patients and 81% in ctDNA-negative patients, respectively.

Similar findings can also be observed for DFS and OS: the possibility of reaching DFS for patients with negative test results was 76%, while in the ctDNA-positive group it was 50%. Considering the overall survival (OS), the 2-year OS rates were 80% and 57% in ctDNA-positive and -negative patients, respectively.

This study also analyzed the effect of two other factors, namely CTC positivity and CTC count. In the case of CTC positivity, the authors did not observe any statistically significant difference between CTC-positive and CTC-negative patients. However, similar to what was reported by Bidard et al. in the metastatic setting, an increase in CTC count had a significant association with worse results in terms of DFS, DDFS, and OS. It is important to note that out of 112 patients for whom both the CTC and ctDNA results were available, no association was found between ctDNA positivity and CTC positivity.

They subsequently divided patients into four subgroups, according to ctDNA and CTC, reporting that the group with positive results for both ctDNA and CTC had significantly inferior outcomes regarding DDFS. Considering the probability of DDFS, the probability of DDFS in the ctDNA- and CTC-positive group was 52% in comparison to 89% for the group that was negative for both criteria. Similar trends were seen when DFS and OS were considered.

Their final results indicated that the combination of ctDNA and CTC can be used as a sensitive prognostic marker in order to predict response to NACT and the survival of the TNBC patients [33].

Finally, Cavallone et al. conducted a study on 26 patients with TNBC, collecting blood samples before, during, and after neoadjuvant chemotherapy, to assess the prognostic and predictive value of ctDNA to NACT. They observed that after start of chemotherapy, a substantial decrease in the ctDNA level occurred, suggesting that this decrease was due to the effect of chemotherapy on the primary tumor’s ability to proliferate and generate ctDNA [34].

Table 3 summarizes the main data obtained with liquid biopsy as predictive factors of response.

#### 3.2.3. Platelet–Lymphocyte Ratio (PLR) and Neutrophil–Lymphocyte Ratio (NLR)

The availability of peripheral blood samples, and their ease of use in terms of rapid access and analysis, allows us to consider two other different immune-related predictive tools in patients with TNBC. Platelet to lymphocyte ratio (PLR) and neutrophil to lymphocyte ratio (NLR) are factors that could be helpful in understanding how patients could respond to the treatment they will receive.

Suee Lee et al. evaluated the prognostic significance of NLR and PLR in patients with gastric cancer treated with chemotherapy: they showed that higher NLR and PLR values were associated with worser outcomes than normal values [35]. These results were consistent with another study that was conducted in 2011 by Jung MR et al. [36].

Similar results have also been obtained in different studies in other cancers, such as those reported by Kwon et al. in patients with colorectal cancer [37].

Three other studies investigated the same factors in the prognosis of lung, ovarian, and pancreatic cancer, all indicating poorer prognosis in patients with higher NLR and PLR [38,39,40].

Yuka Asano et al. investigated the relationship between PLR and pCR in 177 breast cancer patients undergoing neoadjuvant chemotherapy with 5-Fluorouracil, epirubicin, and cyclophosphamide followed by weekly paclitaxel treatment. A low PLR value was detected in a greater number of patients over 56 years old and post-menopausal women; also, this group had higher pCR and longer disease-free survival rate (DFS) in comparison to the high PLR group [41].

In another study, Sejdi Lusho et al. reported a favorable outcome in terms of pCR (yT0/is N0) or small residual disease (ypT1a/bN0) when investigating the role of PLR and NLR. Symmans WF et al. demonstrated that both residual cancer burden 0 (RCB 0) and RCB 1 show a similar trend in relapse-free survival (RFS) 5 years after diagnosis of the disease [42]. Regarding RCB-minimal and RCB-0, Sejdi Lusho et al. demonstrated that the only relevant predictive biomarker is PLR, and that a lower PLR value can be associated with better prognosis. S. Chae et al. reported similar results in patients with TNBC at stages I–III, considering NLR as a biomarker for pCR achievement, meaning that patients with lower NLR could reach pCR in higher numbers than patients with higher NLR [43].

## 4. Discussion

In addition to the importance of having predictive tools to predict pathological response in all molecular subtypes of breast cancer, predicting pCR is of paramount importance in TNBC patients especially, as their prognosis is severe and only a few drugs have demonstrated significant activity. In the neoadjuvant setting, the achievement of pathological complete response (pCR) remains the most important prognostic factor; over the last few years, and with more and more active chemotherapy and targeted therapy regimens becoming available, pCR has become an important surrogate endpoint for predicting long-term clinical benefit, such as disease-free survival, event-free survival (EFS), and overall survival (OS) [44]. In a meta-analysis evaluating 12 international trials with more than 10,000 patients, Cortazar et al. demonstrated that patients who achieve pCR, defined as ypT0 ypN0 or ypT0/is ypN0, have improved outcomes. The prognostic value was greatest in aggressive tumor subtypes, such as TNBC.

Several clinical and biomarker tools have been explored to predict the rate of pCR achievement [45]: stromal tumor-infiltrating lymphocytes (sTILs), programmed-death ligand-1 status (PD-L1), and certain gene signatures, among others, have shown promising but inconclusive results. Currently, two different categories of biomarkers have been retrospectively or prospectively tested in TNBC patients during NACT with or without ICIs: those obtained from the tumor and surrounding tissue (sTILs; RNA disruption assay—RNA-Dx) and those derived from liquid biopsy (ctDNA, among others).

As described in our Results section, sTILs are positively associated with pCR and both sTIL and it-TILs are associated with better prognosis at least in one of the analyzed studies. One of the main limitations of a routine use of TILs as a predictive factor is the lack of prospective trials validating all the encouraging results described by different retrospective trials. Despite this shortcoming, TIL evaluation demonstrates important advantages, as (1) TILs have been validated in terms of cut-off and subsequently endorsed and standardized by the most important guidelines; (2) they are easy to evaluate on both pre-surgical and surgical specimens; and finally, (3) their evaluation is relatively cheap. Some questions, however, remain unanswered:-When should TILs be measured? Definitive results about the prospective use of baseline TILs values are still lacking.-How can TIL variation be monitored? It is still not yet clear when, during the course of neoadjuvant treatment, a second biopsy should be performed to allow for a correct and proper evaluation, as well as how this evaluation correlates with main outcomes.

Among the potential future biomarkers, the burden of TP53 mutation seems to be very promising, but the lack of well-conducted retrospective studies, and prospective trials, limits any further consideration.

Ki67 has been extensively described in different studies: it is well established that a reduction in Ki67 expression at first evaluation in comparison to baseline is strongly correlated with favorable outcomes, in terms of both pCR and survival endpoints. Although, even for Ki67, some questions remain unanswered:-Optimal cut-off values still lacks validation.-An evaluation of how it is associated with other tissue biomarkers is completely absent in the literature and probably deserves more of a focus.

Finally, another important biomarker that should be considered for future research is the presence of circulating tumor cells (CTCs) or DNA fragments released by the tumor (ctDNA, cfDNA), detectable with liquid biopsy. Liquid biopsy is a technique used to monitor drug response in solid tumors. Extensive studies have been conducted on circulating DNA (ctDNA) in breast cancer (BC) patients. Several of them have proposed ctDNA as a prognostic and surrogate marker in BC patients; however, only a few studies have evaluated ctDNA in the neoadjuvant setting [34], and this remains the main limitation regarding routine use.

Very recently, immune checkpoint inhibitors (ICIs) combined with NAC have become the standard of care in stage II-III TNBC patients [46,47,48,49,50,51]. Several studies evaluated factors that could be effective in predicting the response to NACT + ICIs in TNBC patients.

The KEYNOTE-522 trial represented a turning point for the use of ICIs as a neoadjuvant treatment of TNBC: for the first time, a significant improvement in pCR rates was demonstrated by adding Pembrolizumab to standard chemotherapy, impacting the achieved outcomes for TNBC patients. The percentage of patients with a pathological complete response was 64.8% in the Pembrolizumab–chemotherapy group and 51.2% in the placebo–chemotherapy group, with an estimated difference of 13.6 percentage points [47]. Moreover, Pembrolizumab demonstrated a shift in patients, placing them all into lower RCB categories across the entire spectrum compared with the placebo [48]. Finally, Pembrolizumab demonstrated an increase in both event-free (EFS) and overall survival (OS) [49]. Several questions arise after viewing the KN-522 results, among which the most important, from a clinical point of view, are (1) is the adjuvant therapy needed in all patients? (2) Are we able to predict pCR, thus avoiding unnecessary treatment for patients in certain categories? (3) How can we improve this regimen to achieve lower RCB 3?

The results of the KN-522 trial clearly indicated the importance of further research on the application of ICIs in breast cancer treatment, as well as the need to understand underlying mechanisms and factors determining response to treatment.

Several other study results are now available regarding the importance of ICIs as part of the neoadjuvant treatment of TNBC patients.

Table 4 summarizes the key information regarding the above-mentioned trials.

## 5. Conclusions

Despite many developments in the research on predictive biomarkers regarding the response to NAC in TNBC patients, considerable progress is needed to solve this topic. Among the tissue-based biomarkers, sTILs seem to be the most promising as predictive factors in TNBC patients. Among the liquid biopsy-based biomarkers, ctDNA, though promising when analyzed with more modern techniques, failed to demonstrate a robust role as a predictive biomarker of response.

Some of the biomarkers described in this review are still actively under investigation; however, together with considering the scientific needs of the community, we should keep in mind that future biomarker research must be affordable and reproducible in all countries and in all patient populations. Considering the potential cost-effectiveness of such approaches, a careful evaluation of indirect and direct costs is also required in the short, medium, and long term.

## 6. Future Directions

The discovery of immune inhibitory mechanisms, such as CTLA-4, PD-1, and anti PDL-1, lead to a revolutionary understanding of cancer immune evasion, as well as autoimmune diseases. However, the agents that work via the blockage of these inhibitory mechanisms impose significant financial burdens on healthcare systems internationally, leading to some countries not being able to use these agents in patients’ therapy plans and relying on old chemotherapeutic methods. However, more research focusing on the factors affecting the response to ICIs can lead to a more comprehensive understanding of the disease’s biology and the modification of treatment plans based on each patient’s need, leading to unnecessary expenses being curtailed.

Our experience in lung cancer has enabled us to understand that in spite of the fact that anti-PD-1 agents block the interaction of this receptor with its ligands, PD-L1 and PD-L2, PD-L1’s level of expression does not imply a definitive pattern of response to ICIs in these patients, as several studies propose the use of agents such as Nivolumab, Durvalumab, or Pembrolizumab in perioperative settings in non-small-cell lung cancers (NSCLCs) with a PD-L1 expression lower than 1%, reportedly improving the outcomes of patients [53,54,55]. This can indicate the need for further research in order to understand if measuring PD-L1 in a TNBC patient can be useful or not; also, different types of lymphocyte infiltration might modify the approach taken for the treatment, which is another issue that needs further investigation.

Although it is important to investigate tumor behavior in ICI therapy, the chemotherapeutic agents that are being used and the tumors’ response to these agents also needs further evaluation. Considering the increased risk of acute and delayed toxicites in association with certain chemotherapeutic agents, or peripheral neuropathies seen in therapies with taxanes, we need further research in order to provide us with an understanding of each tumor’s behavior against each specific type of these agents, allowing us to avoid any agents which could cause further complications.

## Figures and Tables

**Table 1 curroncol-32-00387-t001:** TILs as predictive factors of response in early BC patients.

Study	Type of Study	Staging	pCR		5-Year Overall Survival		Type of Breast Cancer (BC)	DFS	Cut-Off Definition for High TILs
Denkert et al. (2009) [11]	Cohort		High TILs	Low TILs	NR		all	NR	-
		T2-4; N0-3; M0	42% (training cohort) 40% (validation cohort)	3% (training cohort) 7% (validation cohort)					
Dieci et al. (2013) [12]	Retrospective multicenter	I-II-III- unknown	NR		High TILs	Low TILs	TNBC	NR	60% (either it-TILs or str-TILs)
					91%	55%			
Asano et al. (2018) [13]	Retrospective		High TILs	Low TILs	NR		TNBC HER2BC HRBC	NR	10%
		IIA, IIB, IIIA	46 (26 TNBC; 9 HER2BC; 4 HRBC)	21 (2 TNBC; 9 HER2BC 17 HRBC)					
Gao et al. (2020) [14]	Systematic review and meta-analysis	-	NR		NR		TNBC	High TILs vs. Low TILs	Different in each type of study
								Hazard ratio (HR) = 0.66%; 95% CI = 0.57–0.76	

(TILs = tumor infiltrating lymphocyte; it-TILs = intra-tumoral TILs; str-TILs = stromal TILs; HER2BC = HER-2-positive breast cancer; HRBC = hormone receptor breast cancer; TNBC = triple-negative breast cancer; CI = confidence interval; HR = hazard ratio; NR = not reported).

**Table 2 curroncol-32-00387-t002:** Summary of data regarding Ki67 role as biomarkers.

Study	Type and Stage of BC	NACT Regimen	Cut-Off of Ki-67	pCR
Wang et al.; 2016 [23]	280 TNBC patients; Stage: II–III	Paclitaxel–Carboplatin	Low: <20% Median: 20–50% High: >50%	Low Ki-67: 14.1% Median Ki-67: 29.4% High Ki-67: 58.3%
Keam et al.; 2011 [24]	105 TNBC patients; Stage: II–III	Docetaxel–Doxorubicin	≥10% High Ki-67	Ki-67 ≥ 10% → pCR: 13.3% Ki-67 < 10% → pCR: 0
Raise et al.; 2024 [26]	187 BC patients	Anthracycline based + Taxane	>35%	pCR = 75 patients, out of whom 64 had high Ki-67
Masuda et al.; 2011 [27]	33 TNBC patients; any T + N0-N2	Epirubicin + Cyclophosphamide + Taxane	≥50% → High <50% → Low	High Ki-67 → pCR = 50%. Low Ki-67 → pCR = 15%.

**Table 3 curroncol-32-00387-t003:** CTC and ctDNA as predictive factors of response to neoadjuvant treatment in early BC patients.

Study	Stage of the Disease	Assessed Criteria	DDFS (in 24 Months)	DFS	PFS	OS
Cristofanilli et al. (2004) [30]	metastatic breast cancer (mBC)	Baseline CTC * 5/7.5 mL	-	-	2.7 (months)	10.1 (months)
	metastatic breast cancer (mBC)	Baseline CTC < 5/7.5 mL	-	-	7 (months)	18 (months)
Bidard FC et al. (2014) [31]	metastatic breast cancer (mBC)	Baseline CTC * 5/7.5 mL vs. CTC < 5/7.5 mL	-	-	HR = 1.92 (95%CI = 1.73–2.14)	HR = 2.78 (95%CI = 2.42–3.19)
YH et al. (2014) [32]	early-stage TNBC	ctDNA-positive	-	In 10 months	-	-
				0		
	early-stage TNBC	ctDNA-negative	-	In 10 months	-	-
				>75%		
Radovich et al. (2020) [33]	early-stage TNBC	ctDNA-positive	56%	24 months	-	24 months
				50%		57%
	early-stage TNBC	ctDNA-negative	81%	76%	-	80%
	early-stage TNBC	ctDNA-positive, CTC-positive	52%	54%	-	50%
	early-stage TNBC	ctDNA-negative; CTC-negative	89%	80%	-	85%

(DDFS = distant disease-free survival; DFS = disease-free survival; PFS = progression-free survival; OS = overall survival; ctDNA = circulating tumor DNA; CTC = circulating tumor cell; HR = hazard ratio; CI = confidence interval, * number of cells per each 7.5ml of blood).

**Table 4 curroncol-32-00387-t004:** Summary of the main results of regimens combining chemotherapy with immune checkpoint inhibitors.

Study	Treatment	Population	pCR Rates	3-yr EFS (%)	OS (%)
KN-522 (Schmid, 2020; Schmid, 2022; Schmid, 2024) [46,47,49]	Taxol + Carbo → EC vs. Taxol + Carbo + Pembro → EC + Pembro	Stage II–III	51.2% vs. 64.8%	76.8% vs. 84.5%	81.7% vs. 86.6%
GeparNuevo (Loibl, 2019; Loibl, 2022) [50,51]	Nab-Paclitaxel → EC vs. Nab-Paclitaxel + Durvalumab → EC + Durvalumab	cT1b-cT4a Any N	53% vs. 44%	77.2% vs. 85.6%	83.5% vs. 95.2%
neoTRIP (Gianni, 2022) [52]	Nab-Paclitaxel vs. Nab-Paclitaxel + Atezolizumab	cT1c-cT4d N1 if cT1c or ant N in other T stages	44.4% vs. 48.6%	Not yet reported	Not yet reported

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
