# Peer review of "Predictive Factors of Response to Neoadjuvant Chemotherapy (NACT) and Immune Checkpoint Inhibitors in Early-Stage Triple-Negative Breast Cancer Patients (TNBC)"

_curroncol, 2025, doi:10.3390/curroncol32070387_

Round 1

Reviewer 1 Report

Comments and Suggestions for Authors

In this review, the authors aim to determine factors that may benefit predicting the response to neoadjuvant chemotherapy (NACT) and immune checkpoint inhibitors (ICI) therapies in early-stage TNBC patients. However, the current version of this manuscript is not well structured, and more effort needs to be made to document previous studies in a related area. Some comments or concerns are listed below.

Comments

  1. The abstract needs to be rewritten as it fails to summarize all the essential topics in this manuscript. For example, the authors mainly focus on TILs, genetic alterations (TP53), and CTC/ctDNA.
  2. In the introduction, the authors listed several NACT and ICI studies/trials with tumor types and clinical outcomes (pCR or OS). It would be more straightforward if the authors could summarize this information in a table. Furthermore, the authors should highlight a new perspective or relatively specific focus, such as liquid biomarker ctDNA.
  3. This manuscript focuses on early TNBC patients. The authors should include descriptions of tumor stages or grade information related to specific studies or trials.
Comments on the Quality of English Language

The manuscript has grammatical errors. An English native-speaking person should go through and correct these errors.

Author Response

we fully answered point by point in the attached file, as the comments are too long for this space

Reviewer 2 Report

Comments and Suggestions for Authors

The authors have submitted a valuable and timely review on predictive markers for early chemotherapy and immune checkpoint inhibitor use in triple-negative breast cancer. This is an important area of focus and the authors do a good job of summarizing literature in this area including recent reports. The authors also supplement the manuscript with good insight on the applicability of using this information in the clinical setting.

Addressing the points below will help to improve the overall quality of the article.

The manuscript would benefit from an English editing service. There are issues with sentence structure, capitalization, phrasing and repetitive instances of term definitions throughout the manuscript.

In regard to the title, while ICIs are briefly mentioned in the introduction, there is minimal or no mention of ICIs in the three major sections of the discussion (TILs, p53, liquid biopsies). The studies by Loibl et al involving durvalumab referenced in the introduction should be greatly expanded in detail as these are some of the few studies included that directly relate to the theme of the article, especially the GeparNuevo study.

Additional information regarding the KEYNOTE-522 study would also be helpful. For example, a pCR rate of 64.8% is listed for the pembrolizumab + chemo group but there is no mention of the pCR rate in the chemo only group (51.2%) to provide context for the reader.

It would be beneficial to include an additional column for Table 2 indicating the type of neoadjuvant therapy for each study.

Discussion of studies evaluating Ki-67 as a predictive marker for neoadjuvant chemotherapy may be helpful to include, especially as it relates to p53 status. There are a few references already included by the authors related to this (Pan et al, Faur et al).

Author Response

(The authors gave the same response as above.)

Reviewer 3 Report

Comments and Suggestions for Authors

The authors present a well-organized and timely review focused on predictive biomarkers influencing pathologic complete response (pCR) to neoadjuvant chemotherapy in triple-negative breast cancer (TNBC). The topic is clinically important, and the manuscript covers relevant biomarkers such as TILs, ctDNA, CTCs, and TP53 mutations in detail. The introduction lays a solid foundation for the review by discussing TNBC’s aggressive nature and the role of NACT and immune checkpoint inhibitors. The literature survey appears extensive and well-referenced. However, several aspects could be improved to enhance the clarity and overall impact of the manuscript:

  • Introduction: The authors should clearly explain why pCR is a crucial surrogate endpoint in TNBC and how it informs prognosis and treatment planning. Additionally, a concise statement on the specific research gap this review addresses would strengthen the framing of the paper.
  • Biomarker Selection: It would help to briefly explain why certain biomarkers (NLR, PLR, TP53) were chosen for discussion, especially considering the broad landscape of potential predictive markers.
  • Methods: The process of narrowing down from 77 to 43 studies lacks detail. Including basic inclusion/exclusion criteria or a short explanation of the filtering process would improve transparency and reproducibility.
  • Discussion: Some terms like “predictive” and “prognostic” biomarkers are used interchangeably; defining them early and maintaining consistent usage throughout would improve clarity. Additionally, some transitions between paragraphs feel abrupt and could be better connected. A brief mechanistic rationale for how biomarkers like NLR and PLR relate specifically to TNBC treatment response would also be useful.
  • Conclusion: This section is currently quite generic. It should more clearly summarize the key findings from the review and outline concrete future directions or unanswered questions in the field.
  • Language and Formatting: While the review literature is strong, the manuscript would benefit from a round of scientific language editing to improve flow and clarity (e.g., lines 84, 204, 223, 262, 423). There are also scattered formatting issues with references and citation colors that should be cleaned up.
Comments on the Quality of English Language

The language is too generic - some sentences sounds like two persons talking to each other. Please consider re-writing that will sound scientific.

Author Response

(The authors gave the same response as above.)

Round 2

Reviewer 2 Report

Comments and Suggestions for Authors

The authors have addressed reviewer concerns and made appropriate revisions including the addition of new content and sections.

Author Response

Thank you for your comments.

Reviewer 3 Report

Comments and Suggestions for Authors

The authors have done great job addressing previous comments thoroughly. Additional markers like Ki-67 and ICI's have been included in revised review, adding relevance to the article. The overall clarity and flow was improved after english editing. I have no further suggestions and recommends for acceptance.

Author Response

Thank you for your comments.